# Urban Sprawl, Food Subsidies and Power Lines: An Ecological Trap for Large Frugivorous Bats in Sri Lanka?

**José L. Tella [1],\*, Dailos Hernández-Brito [1] , Guillermo Blanco [2] and Fernando Hiraldo [1]**

1   Department of Conservation Biology, Estación Biológica de Doñana (CSIC), Avda. Américo Vespucio 26, 41092 Sevilla, Spain; dailoshb@gmail.com (D.H.-B.); hiraldo@ebd.csic.es (F.H.)
2   Department of Evolutionary Ecology, Museo Nacional de Ciencias Naturales (CSIC), José Gutiérrez Abascal 2, 28006 Madrid, Spain; g.blanco@csic.es
*   Correspondence: tella@ebd.csic.es

**Abstract:** Electrocution is one of the less known anthropogenic impacts likely affecting the bat population. We surveyed 925 km of overhead distribution power lines that supply energy to spreading urbanized areas in Sri Lanka, recording 300 electrocuted Indian flying foxes (*Pteropus giganteus*). Electrocutions were recorded up to 58 km from the nearest known colony, and all of them were in urbanized areas and very close ($\overline{X}$ = 4.8 m) to the exotic fruiting trees cultivated in gardens. Predictable anthropogenic food subsidies, in the form of cultivated fruits and flowers, seem to attract flying foxes to urban habitats, which in turn become ecological traps given their high electrocution risk. However, electrocution rates greatly varied among the 352 power lines surveyed (0.00–24.6 indiv./km), being highest in power lines with four wires oriented vertically ($\overline{X}$ = 0.92 indiv./km) and almost zero in power lines with wires oriented horizontally. Therefore, the latter design should be applied to projected new power lines and old vertically oriented lines in electrocution hotspots should be substituted. Given that flying foxes are key seed dispersers and pollinators, their foraging habitat selection change toward urban habitats together with high electrocution risk not only may contribute to their population decline but also put their ecosystem services at risk.

**Keywords:** anthropogenic food subsidies; ecological traps; ecosystem services; electrocution; exotic plants; fruit bats; power lines; seed dispersal; urbanization

## 1. Introduction

Bats (Order Chiroptera) stand out among mammals by their high diversity, with c. 1400 species widely distributed across the world, and by their poor conservation status and little knowledge on their ecology and conservation threats [1]. According to IUCN, 15% of bat species are threatened with extinction, 18% are listed as Data Deficient, and 57% have unknown population trends, figures that are higher than for any other mammalian groups and birds as a whole [1]. Although much more research is needed, a number of human-induced impacts on bat population have long been identified, such as habitat loss, disturbance, persecution or overexploitation for food [1–3], while others have been highlighted more recently, such as light pollution [4] or competition with introduced invasive species [5]. Moreover, new conservation threats related to the increasing energy demand worldwide have emerged in recent years. In particular, bat fatalities at wind power facilities are attracting the attention of researchers [6–8], as they may cause the death of millions of bats a year across the world [3]. However, very little is known about the potential impact of electrocution at distribution and transmission power lines on bat populations [9].

Electrocution usually occurs when an animal makes contact with two energized wires or an energized wire and grounded structure [10]. While large-bodied birds are usually highly vulnerable, the small size of most bat species reduces their risk of electrocution, as it seems to occur in the United States where bats have been found incidentally in bird mortality searches in powerline corridors [10]. However, large-bodied tropical bats, such as flying foxes (*Pteropodidae*), may be at higher risk since their large wingspan (often >1 m) may facilitate the contact with two or more energized wires. In fact, there is a number of supposedly anecdotal cases of different flying fox species fallen victim to electrocution because of power lines [9], which often are reported in gray literature [11–19]. Nonetheless, there are two cases of great concern. First, two monitoring studies of individually banded grey-headed flying-foxes (*Pteropus poliocephalus*) in Australia showed that electrocution at power lines accounted for 71% and 18% of the dead individuals recovered [20,21]. Second, there is a report of 74 Indian flying foxes (*Pteropus giganteus*) found electrocuted over just a 3-km stretch of power line at the Paradeniya Botanic Garden, Sri Lanka [22]. Doubts arise, however, about the actual impact of these fatalities at large spatial and population scales since these high electrocution rates could result from particularities at a local scale, such as the close proximity of a flying fox colony or the presence of an unusually dangerous power line.

Here, we present the results of a large-scale baseline survey of overhead distribution power lines (925 km) conducted in Sri Lanka to assess electrocution risks for the frugivorous Indian flying fox (*P. giganteus*), which is one of the largest bat species of the world (weight: 0.8–1.4 kg, wingspan: c. 1.2 m [23]) (Figure 1a,b). This species roosts in large colonies of hundreds to thousands of individuals in large trees (Figure 1a), leaving them at night (Figure 1b) to travel long distances, up to 150 km, in search of fruits and flowers [24]. We related the spatial distribution of electrocuted flying foxes (Figure 1c) to the proximity of colonies, the main habitats and degree of urbanization, and the presence of fruiting trees. Although mortality rates varied between power line types (Figure 1d,e), they were always found in highly urbanized areas and very close to cultivated exotic fruiting trees. Our results suggest that the widespread urban sprawl in Sri Lanka initially favors flying foxes by providing them with predictable anthropogenic food subsidies (fruits and flowers), but turns into an ecological trap given the high mortality rates caused by the power lines needed to supply the homes with energy. We discuss the implications on the conservation of this species in this and other areas of distribution, as well as the loss of their ecological functions (seed dispersal and pollination), and the potential of different mitigation measures to reduce mortality rates.

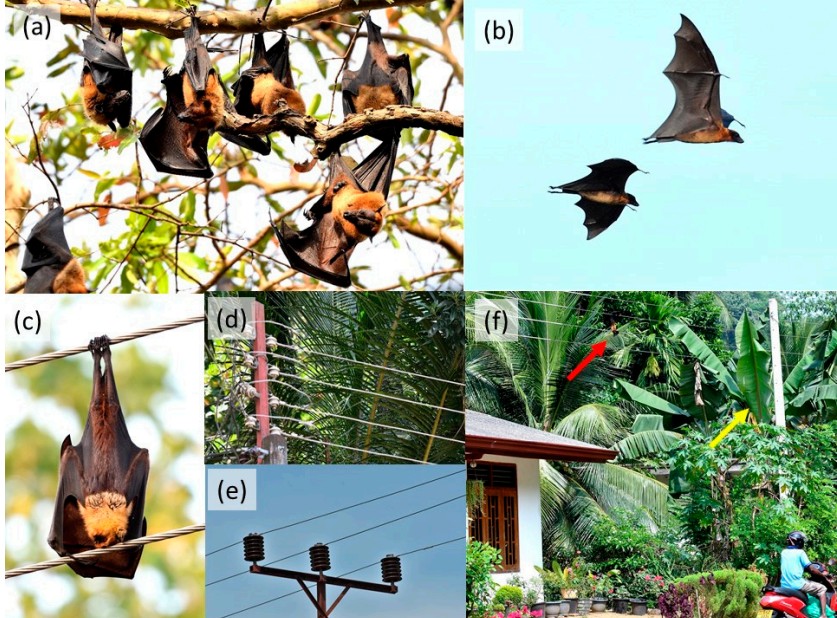

**Figure 1.** Indian flying foxes gather in diurnal colonies (**a**) and leave them at sunset (**b**) to forage on fruit trees. They were frequently found electrocuted when perching on wires (**c**) of two different types of distribution power lines (**d**,**e**), always in urbanized areas (red arrow) and very close to cultivated fruiting trees (yellow arrow) (**f**). Pictures: J.L. Tella.

## 2. Materials and Methods

Fieldwork was conducted between 10th and 28th January 2018 in the southern half of Sri Lanka (Figure 2). We drove a number of primary and secondary paved and unpaved roads across the study area (Figure 2) looking for overhead distribution, medium-voltage power lines that supply energy to and between human settlements. We recorded all the Indian flying foxes found electrocuted in those power lines (*n* = 300 individuals) during the roadside survey to explore the spatial distribution of fatalities (see below). However, to estimate mortality rates (number of electrocuted individuals/km) related to power line types, we only used those individuals (*n* = 266) found in power lines from which we clearly identified where they start and where they finish, and that could be fully surveyed from the car. This resulted in 351 fully surveyed power lines. We measured their length using a portable GPS while driving the car, averaging 2.63 km per power line (range: 0.02–27.67 km) and summing a total of 925.25 km of power lines. We recorded three main types of medium-voltage power lines: those where wires are oriented vertically (Figure 1d), with three or four wires (thereafter types 3V and 4V, respectively) and those where wires are oriented horizontally (Figure 1e, thereafter type H). Type 4V was the most commonly found (160 power lines, 482.80 km), closely followed by H (153 power lines, 398.86 km), while 3V power lines were scarce (38 power lines, 43.59 km). We visually classified the landscape surrounding each power line into three main habitat categories: (1) urban (i.e., streets of large cities), (2) suburban (villages and landscapes dominated by small human settlements and scattered buildings), and (3) agricultural (small human settlements embedded into an agricultural matrix). Most power lines were located in suburban areas (281 power lines, 793.91 km), followed by urban (52 power lines, 70.48 km) and agricultural areas (18 power lines, 60.86 km).

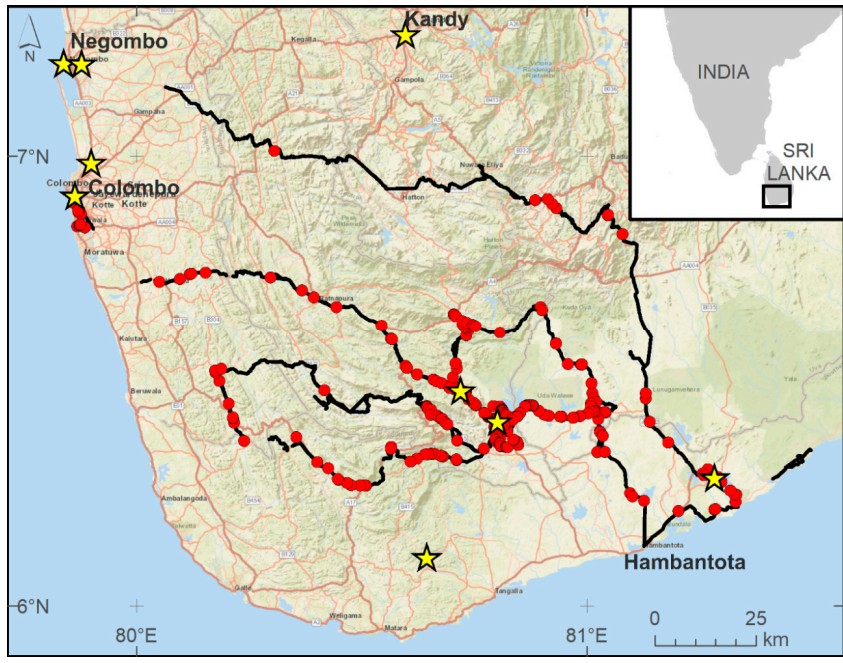

**Figure 2.** Study area in the Southern half of Sri Lanka. Black lines are the roads driven to locate distribution power lines and electrocuted Indian flying foxes. Yellow stars show the colonies known of this species, and red circles the location of electrocuted individuals.

All the power lines surveyed were along roadsides and streets, running in parallel and very close (usually 2–5 m) to the trajectory of the car. Therefore, the big-sized, electrocuted flying foxes hanging from wires (Figure 1c) were easily detectable by four people (three observers and the driver) while driving the car at low speed (20–40 km/h). There was no evidence for differences in detectability among habitats and power line types. Every electrocuted individual was recorded using the application ObsMapp for smartphones, which uploads the observations to the citizen science platform Observation (www.observation.org) [25,26]. Therefore, all electrocution records, exact location, and associated information can be viewed and downloaded from this platform, and it may engage other people to add further electrocution records in the future [25,26].

During the course of fieldwork, we noted that electrocuted individuals were close to cultivated fruiting trees. Consequently, after January 17 we also systematically recorded the number and species of fruiting trees in a 30 m radius around each electrocuted individual as well as the distance to the nearest fruiting tree. Therefore, this kind of information is available for a good subsample of all the fatalities recorded (*n* = 102; 15 in urban, 83 in suburban, and 4 in agriculture habitats). Distances <8 m were estimated visually or by counting paces, while a laser rangefinder incorporated into binoculars (Leica Geovid 10 × 42) was used for measuring the larger distances. During the field surveys we also looked for colonies of flying foxes (i.e., trees where the species gathers in high numbers to roost during daylight hours, Figure 1a), also utilizing the information available in Observation and that provided by local people. Colonies are highly conspicuous and well-known by people, as they concentrate hundreds to thousands of individuals and are used year around and across years [24]. We also recorded the location of colonies with ObsMapp and estimated their size following previous studies [16,22,27,28], by simultaneously counting the number of individuals hanging from the tree branches with the help of binoculars (four people counting the bats simultaneously in different parts of each colony).

Once in the lab, we downloaded from Observation all georeferenced locations of electrocuted individuals and colonies and plotted them in Google Earth. Then, as measures of urbanization, we calculated the distance from each point to the nearest building as well as the number of building within a 100 m buffer. We also measured the straight-line distance from each electrocuted individual

to the nearest known colony. We used ARCGIS 10.2.2 (Environmental Systems Research Institute, Redlands CA) for spatial data management and calculations of distances.

The distribution of mortality rates (number of individuals electrocuted/km) clearly followed a Poisson distribution. Therefore, for assessing sources of variation in mortality rates among power lines, we built a generalized linear model (GLM) for count data, with a Poisson distribution and log link function. The number of electrocuted individuals counted at each power line was the response variable, fitting the length of the power line as a covariate as the number of fatalities is expected to increase with power line length. As explanatory variables, we fitted type of power line and habitat as fixed factors. In a number of occasions, two different types of power line coincided in space (i.e., one type was on the left side of the road and another type was on the right one, $n = 166$ for H and 4V types and $n = 16$ for H and 3V types). This offered a unique scenario, like a "natural" experiment, for testing the differences in mortality rates exclusively attributed to the power line type, as any other unmeasured effect associated with the particular location of the power line is controlled for. Therefore, we repeated the GLM retaining only those power lines that coincided in space with others. Analyses were performed using SPSS v. 25.0.

## 3. Results

### *3.1. Location of Electrocuted Flying Foxes*

We recorded 300 flying foxes electrocuted in low-voltage power lines and located nine colonies within the same study area (Figure 2), each comprised of between 50 and 9250 individuals (summing a minimum of 19,824 individuals, as one colony could be not surveyed). Electrocuted individuals were found at an average distance of 15.42 km (SD = 15.12, range = 0.069–57.93 km) from the nearest known colony, and most of them (83.7%) were within a radius of 25 km (Figure 3a). All fatalities were recorded within or very close to human settlements. There was an average of 14.93 buildings (SD = 16.11, range = 1–115) within a radius of 100 m around each electrocuted individual (Figure 3b), and the mean distance to the nearest building was 27.08 m (SD = 13.91, range = 2–81 m; 75.3% within 30 m; Figure 3c). All colonies were in urbanized areas, with an average of 15.8 buildings within 100 m (SD = 21.34, range: 1–59) and a mean distance of 40.56 m to the nearest building (SD = 30.67, range: 8–91 m).

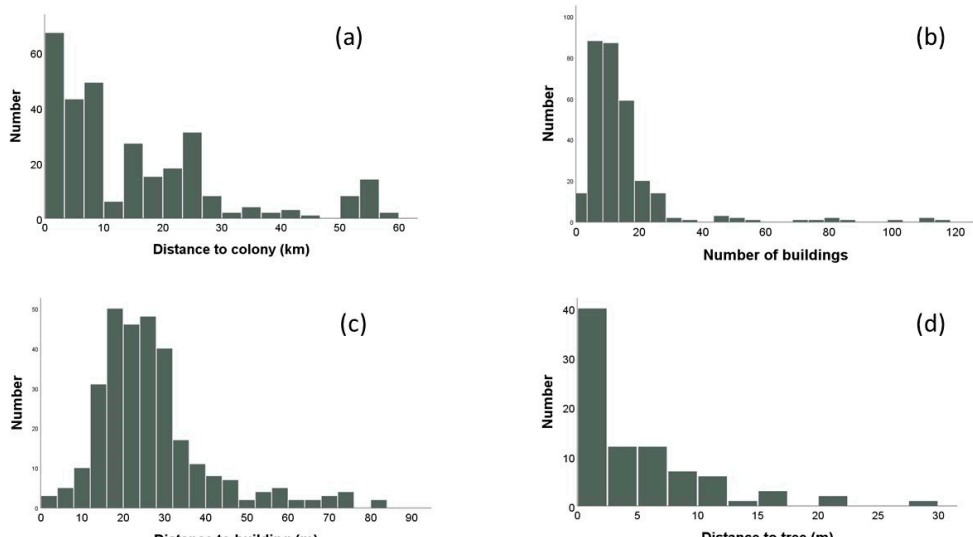

**Figure 3.** Number of flying foxes found electrocuted in relation to (**a**) their distance to the nearest known colony, (**b**) the number of buildings within a 100 m radius, (**c**) the distance to the nearest building, and (**d**) to the nearest cultivated fruiting tree.

For a sample of 102 electrocuted bats, we recorded at least one cultivated fruiting tree (range = 1–3 trees) within a radius of 30 m in all but in two cases. Fruiting trees corresponded to 13 species, from which one was a native species (*Limonia acidissima*), ten were exotic species, and one was a cultivated hybrid (Table 1). Exotic and hybrid trees summed 99.2% of the individual trees recorded (*n* = 122, Table 1). The average distance from electrocuted bats to the nearest fruiting tree was 4.8 m (SD = 5.6, range = 0–30 m), and most of them (85.9%) were within a radius of 10 m (Figure 3d).

**Table 1.** Number and percentage of cultivated fruiting trees (*n* = 122) located within a 30 m radius of electrocuted flying foxes.

| Tree Species | N | % |
|---|---|---|
| *Mangifera indica* | 51 | 41.80 |
| *Musa x paradisiaca* | 26 | 21.31 |
| *Psidium guajava* | 10 | 8.20 |
| *Artocarpus heterophyllus* | 9 | 7.38 |
| *Ficus religiosa* | 9 | 7.38 |
| *Terminalia catappa* | 6 | 4.92 |
| *Carica papaya* | 4 | 3.28 |
| Unidentified species | 2 | 1.64 |
| *Limonia acidissima* | 1 | 0.82 |
| *Annona muricata* | 1 | 0.82 |
| *Annona reticulata* | 1 | 0.82 |
| *Areca catechu* | 1 | 0.82 |
| *Chrysophyllum monopyrenum* | 1 | 0.82 |

## 3.2. Risk of Electrocution

From the 300 flying foxes found electrocuted, 266 were recorded across the 352 fully surveyed power lines (summing 925.25 km). This rendered a mortality rate of 0.29 electrocuted individuals per km of power line. Mortality rates greatly varied among the 352 power lines fully surveyed, ranging from 0.00 to 24.62 fatalities/km, being higher in 4V (where 94% of the electrocuted individuals were found, averaged means: 0.92 indiv./km) than in 3V (0.74 indiv./km), and being almost null in H (0.01 indiv./km) (Figure 4).

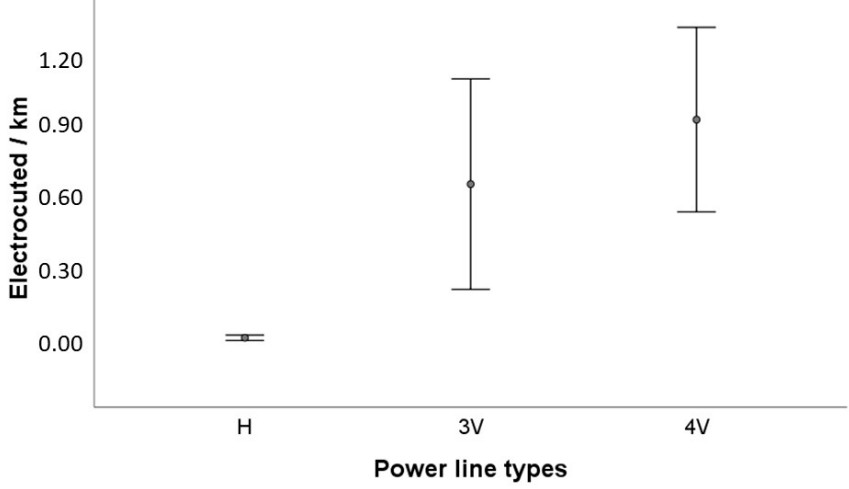

**Figure 4.** Averaged means (and 95% CI) of electrocuted flying foxes per km in 352 power lines according to their three types (H: horizontal wires; 3V: 3 vertical wires; 4V: 4 vertical wires).

Differences in mortality rates between the three kinds of power lines were statistically significant, when also controlling for the length of each power line and for habitat (Table 2). Mortality rates were

higher in urban than in suburban areas with intermediate values in agricultural habitats (Table 2). Results were nearly identical when excluding from analysis the two power lines with extreme cases of mortality (16.67 and 24.62 individuals/km). Differences in mortality rates between 4V and H power lines remained statistically significant when analyses were restricted to sites where both power line types were simultaneously present ($n$ = 166 power lines), one at each side of the road (GLM, power line type effect; Wald $\chi^2$ = 41.30, $p$ < 0.001, habitat effect: Wald $\chi^2$ = 35.46, $p$ < 0.001; length of power line effect: Wald $\chi^2$ = 68.31, $p$ < 0.001). The GLM for the cases where 3V and H power lines coincided in space did not converge because of the small sample size ($n$ = 16 power lines), but still mortality rates were much higher in 3V (0.18 indiv./km) than in H (0.00 indiv./km). Therefore, differences in mortality rates between power line types are not influenced by any particularity associated to their spatial location.

**Table 2.** Results of the generalized linear model for assessing variability in electrocution rates of flying foxes. Explained deviance: 45.61%.

|  | Estimate | s.e. | Wald $\chi^2$ | d.f. | $p$ |
|---|---|---|---|---|---|
| Intercept | 0.865 | 0.1253 | 47.682 | 1 | 0.000 |
| Power line: H | −4.332 | 0.5809 | 55.595 | 1 | 0.000 |
| Power line: 3V | −1.477 | 0.3091 | 22.827 | 1 | 0.000 |
| Power line length | 0.080 | 0.0100 | 63.455 | 1 | 0.000 |
| Habitat: agricultural | −0.525 | 0.2496 | 4.416 | 1 | 0.036 |
| Habitat: suburban | −0.930 | 0.1491 | 38.943 | 1 | 0.000 |

## 4. Discussion

### 4.1. Urbanization as an Ecological Trap for Flying Foxes

While many bat species are negatively affected by urbanization, others may tolerate it or even be favored by its roosting and/or foraging opportunities [29]. This seems to be the case of Indian flying foxes in Sri Lanka. In his field guide book for Sri Lankan mammals, G.D.S. Wijeyeratne stated: "I have never come across a colony of common (Indian) flying foxes in a wooded area remote from human habitation. This may be because foraging in gardens and plantations provides a much higher nutritional yield and foraging efficiency than foraging in native forest. Common flying foxes are not hunted in Sri Lanka and may feel their colonies are more secure near people" [23]. Our large-scale survey supports such observations. On the one hand, all known colonies in our study area are located in urbanized sites, which can constitute predator-free (including the absence of human persecution) refuges as it has been shown for some bird species [30,31] and suggested for a colony of Indian flying foxes in Delhi, India [27]. On the other hand, the electrocuted individuals we found were in urbanized areas and in close proximity to fruiting trees cultivated in gardens. The same proximity to fruit-bearing trees was highlighted for 11 Indian flying foxes found electrocuted in Andaman Islands [15].

There is a variety of ways that humans involuntarily provide food to other animals, constituting predictable anthropogenic food subsidies [32] as it seems to be the case of fruit trees cultivated in urban areas for flying foxes [15,23, this study]. Anthropogenic food subsidies often increase individual fitness and trigger population growth of opportunistic species [32], and thus urban areas have the potential to emerge as "conservation hotspots" [30] for flying foxes. However, the high risk of electrocution for flying foxes that are attracted by cultivated fruiting trees, given the widespread power line grid existing across urbanized areas, may change such a view. Our survey turns previous seemingly anecdotal information on electrocution fatalities in Sri Lanka [22] into a serious source of mortality at a large spatial scale. Therefore, although predictable food subsidies in the form of cultivated fruits and flowers could increase nutritional yield and foraging efficiency [23], and thus individual fitness [32], the high electrocution rates we are reporting may create population sinks and thus turn urban habitats into ecological traps for the species [29].

*4.2. Conservation Implications: Population Decline and the Loss of Ecological Functions*

The Indian flying fox was listed as a Least Concern species in the IUCN Red List in 2008, although recognizing that the population is continuously declining and the conservation status of the species needs an updated reassessment [24]. Electrocution was not considered as a threat to the species at that time [24], but the situation may have changed over the past 12 years. Yet in 2002, B. Krystufek recorded 24.66 electrocuted individuals per km in a single Sri Lankan power line [22], a figure almost identical to the upper range limit of electrocution rates we have found 16 years later for a much larger sample of power lines across an extensive area. Our results are alarming both by the large spatial scale of electrocution risk and by the high number of fatalities. We found 300 electrocuted individuals over just a 19-day survey period, roughly representing 1.5% of the individuals gathering in colonies at the same time. Although we could not survey one colony, neither can we completely discard the presence of additional colonies in the study area, and thus the size of the population is probably higher, the actual number of electrocutions were surely much higher. We surveyed less than one-third of the roads (and thus power lines) crossing and communicating human settlements across the study area (Figure 2). Moreover, an unknown proportion of electrocuted individuals may fall to the ground and could be scavenged by dogs and other domestic or wild animals, thus being overlooked even when conducting ground surveys [22]. A study conducted on a long-term declining colony of Indian flying foxes in Assam, India, indicated that 3–4% of individuals died per night because of electrocution, but no information was provided to support such a high mortality rate [14]. On the other hand, it is likely that electrocutions increased in recent years in Sri Lanka, as a result of the enlarged electricity distribution grid (the on-grid installed electrical capacity increased from < 2000 Mw to > 4000 Mw between 2000 and 2018; [33]) as a response to population growth (from 19 to 22 million people in the same period), resulting in 90% of the households already electrified [34]. It is also expected that electrocution risk will continue to increase, given the current human population and economic growth in Sri Lanka [34]. Contrasting to flying foxes, electrocution risk for birds seems to be low, as we only found one house crow (*Corvus splendens*) electrocuted on vertically oriented wires, despite this species is widespread and very abundant across the study area.

As typically occurs in *Pteropus* species with a slow pace of life [21,35], Indian flying foxes have low fecundity (one young born at year [24]) and any small rates of non-natural adult mortality could seriously impact their demographic trends. In fact, the population modeling of three *Pteropus* species in Australia suggested that two of them were in imminent danger of extinction if current rates of human-induced mortality persist [35]. Therefore, long-term monitoring programs e.g., [36] of flying fox colonies, together with the systematic survey of electrocution rates, should be implemented in Sri Lanka for assessing the actual impact of electrocution on population trends. A major attention should be also paid to other areas within the species distribution where electrocution has been also, although anecdotally, reported [11–18]. In fact, Y.P. Sinha [11] stated that electrocution is a regular cause of death of flying foxes in almost all the cities and states of India. This is an added threat to the conservation of the species elsewhere since, contrarily to what it occurs in Sri Lanka [23], the species is often persecuted because of damage to fruit crops and hunted for meat and medicine [24].

Urban sprawl accompanied with electrocution risk may not only threaten the flying fox population but also affect the ecosystem functioning in Sri Lanka. Indian flying foxes, as other fruit bats, feed on flowers (nectar) and fruits of a variety of tropical trees, and thus are considered as effective pollinators and seed dispersers [28,37–39]. Particularly, they are considered key dispersers of a variety of seeds, from tiny seeds that are swallowed together with the fruit pulp and later defecated at long distances (>10–100 km), to large seeds of big fruits that they move in flight to feeding roosts, where they handle them for consumption and discard the seeds [38]. The large contribution of fruit bats to the structure, biomass, and diversity of native forests has been shown in Mauritius, where the island population of a threatened flying fox (*Pteropus niger*) was reduced >50% by culling to reduce its impact on fruit crops [40]. As it happens with some parrot species [31], the switch of flying foxes from foraging in native forests to urban areas in Sri Lanka [23] may lead to the loss of their ecological functions.

In fact, 99% of the fruiting trees grown in urban gardens that attracted the electrocuted individuals were cultivated exotic or hybrid species. This change in foraging habitat selection, together with the numerical reduction of flying fox population due to electrocution, may mean that their ecological functions might be lost long before the species could become extinct [41]. Recent work suggests the need to consider the ecological functions played by species when assessing their conservation status and recovery plans [42], and thus concerns about the population declines of flying foxes [24] increase when considering the loss of their ecosystem services.

*4.3. Mitigation Options and Further Research*

Knowing how bat fatalities occur and assessing their sources of variation are essential for properly designing mitigation measures. For example, research dealing with bat fatalities at wind energy facilities is growing in recent years and increasing focus has been given on developing effective mitigation and minimization strategies [6–8,43–45]. However, to our knowledge, ours is the first approach dealing with bat electrocutions, from which we can provide some preliminary recommendations.

All electrocuted flying foxes we found were hanging from wires (Figure 1c), suggesting that most of them—if not all—did not die because of collision in flight but when they perched on the power lines and touched two or more energized wires with their body or wings, as it has been previously suggested [11–19]. Large fruit bats usually pick up fruits from trees and transport them to other trees ("feeding roosts") to handle and eat them [38]. The close proximity of electrocuted bats to cultivated fruiting trees suggests that they used wires as substitutes of tree branches for perching and handling the fruits. This makes electrocution a larger spatial problem to solve, since urban sprawl together with tree gardens and small plantations are widespread across Sri Lanka. However, mortality rates varied among power lines from almost no fatalities to c. 25 electrocutions/km, suggesting the existence of mortality hotspots. However, the close proximity to flying fox colonies seems to be not a good predictor of electrocution hotspots. Although the distances we measured could be biased by the existence of some unknown colonies, Figure 3a suggests that electrocutions occur over large distances from them. This may be explained by the fact that flying foxes cover large distances (>100 km) every night for foraging, and even make migratory movements [24]. In fact, one little red flying fox (Pteropus scapulatus) was found electrocuted in Australia 482 km from where it was banded almost two years ago [21]. Therefore, further research is needed in Sri Lanka to identify the electrocution hotspots. Citizen science could help in this way. For instance, focused campaigns using the platform Observation allowed the large-scale spatial monitoring of roadkills in Belgium [26] and of alien invasive species in northwest Europe [25]. Sri Lanka is among the world's most preferred tourism destinations, also for wildlife lovers. Therefore, it should not be difficult to engage people to publicly record electrocutions of flying foxes using their smartphones, once they are made aware of the decreasing population trends and key ecosystem services provided by these animals. It would also improve the poor public perception of bats as a whole [46].

Regarding mitigation options, our results clearly show that distribution power lines with wires oriented horizontally (Figure 1e) cause comparatively little impact, most likely because when flying foxes perch on these wires there are no other wires below that could make contact with them. Therefore, new regulations should impose horizontal wires for newly projected power lines, especially those crossing or connecting nearby urban areas. On the other hand, there are several mitigation possibilities for the already existing power lines with wires oriented vertically (Figure 1d), especially those with four wires (that cause the larger mortality rates) and in those electrocution hotspots that could be identified in the future. Increasing the current distance between wires (c. 30 cm) of the power lines surveyed to c. 1.5 m to avoid electrocution of Indian flying foxes (with a wingspan of 1.2 m) is not realistic, since it would make the construction of overhead medium-voltage power lines impractical [17]. Rather, the insulation of wires, the substitution of vertically, by horizontally oriented wires, or the substitution of overhead power lines by underground electric cables are alternatives that should be considered depending on their economic costs. From an economic point of view, these mitigation costs should be

balanced against those derived from power outages caused every year by the frequent electrocution of flying foxes in the old, non-insulated overhead electrical wiring of Sri Lanka.

Finally, further research is needed to assess the actual impact of electrocution in this and other species of large fruit bats and in other countries. On the one hand, we hope our large-scale baseline survey will trigger more detailed studies to properly estimate daily electrocution rates in Sri Lanka, investigating how long carcasses persist hanging from wires, what proportion of electrocuted individuals fall to the ground, and the potential removal of carcasses by scavengers. Further work should also focus on potential seasonal variation in mortality rates. All this kind of detailed information, together with that obtained from citizen science and adequate statistical tools [47], would help to identify mortality hotspots, which is crucial to optimize resources for mitigating the impact of electrocution. On the other hand, as electrocution seemed until now anecdotal for Indian flying foxes [11–18], the same overlooked threat could occur in other instances. In fact, electrocution has been reported for *P. poicephalus* and *P. scapulatus* in Australia [21], *P. hypomelanus* in Myanmar [17], *P. seychellensis* in Mahe, Seychelles [9], *P. dasymallus* in Japan [48], *Megaerops niphanae* and *Cynopterus sphinx* in India [19], and for an unidentified species in Uganda [49]. Given that >34% of the 184 extant Pteropodidae fruit bat species are threatened with extinction and that 65% of the species for which there is available information (*n* = 124) show decreasing population globally [50], any increase in non-natural mortality caused by electrocution could further contribute to their decline. Therefore, the potential impact of electrocution on other species and countries should be also investigated.

**Author Contributions:** All authors have read and agreed to the published version of the manuscript. Conceptualization, J.L.T., D.H.-B., G.B., and F.H.; methodology, J.L.T., D.H.-B., G.B., and F.H.; statistical analysis, J.L.T.; writing—original draft preparation, J.L.T.; writing—review and editing, J.L.T., D.H.-B., G.B., and F.H.; visualization, J.L.T., D.H.-B., G.B., and F.H.; supervision, J.L.T.; project administration, J.L.T.; funding acquisition, J.L.T.

**Acknowledgments:** We thank D. Hayden, D. Aragonés, and I. Afán for their help managing and obtaining spatial data; to J. Rabadán for his help in using and managing ObsMapp and Observation; and to S. Young for the English editing of the manuscript. Logistic and technical support for fieldwork were provided by Doñana ICTS-RBD, and by LAST-EBD for acquiring spatial information. Three anonymous reviewers helped to improve the original version.

**Conflicts of Interest:** The authors declare no conflict of interest

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
