# Peer review of "Urban Sprawl, Food Subsidies and Power Lines: An Ecological Trap for Large Frugivorous Bats in Sri Lanka?"

_diversity, doi:10.3390/d12030094_

Round 1

Reviewer 1 Report

I have carefully read the article “Urban sprawl, food subsidies, and power lines: an ecological trap for large frugivorous bats in Sri Lanka” submitted by Tella and co-authors. The manuscript is well written and easy to follow, providing important information on the extent to which Indian flying foxes are electrocuted at power lines in southern Sri Lanka. I think the manuscript will be of interest to the readers of Diversity and, if published, may help shed light on the importance of this issue and garner support for implementing mitigation strategies. I have two major comments and several minor questions/comments for the authors.

There needs to be some mention of how detectability of electrocuted flying foxes could vary among the power lines types, as well as among the 3 landscape types, and how this would influence the estimates of electrocution. For example, the authors state that electrocuted flying foxes wouldn’t be found on the ground below horizontal wires, but it appears that the authors did not substantiate this claim (i.e., they did not conduct ground surveys under any wire type). I think the authors’ explanations make sense (that flying foxes get electrocuted when they roost on 4V and 3V wires and therefore can contact 2 wires at once, and that electrocution risk is much less when they roost on H wires), but this should be verified. It would also be helpful to have some idea about how long carcasses persist in the wire and are therefore able to be detected – is it one or two days or can it be much longer? Are the wires equally visible by human observes along the roads? I would imagine that wires could be obscured by buildings in the more urban areas and obscured by trees and other vegetation in the suburban or agricultural areas. Although the authors don’t appear to have data on detectability, this is an important issue that should be addressed in the manuscript.

The authors mention hotspots of fatality as being particularly important, in part for their potential to be strategically mitigated, but their analyses do not appear to reveal any hotspots. In line 182, they state that differences in mortality are not due to any particular spatial location. Is there another analytical approach that could be used to detect “hotspots” in the data? If hotspots cannot be identified quantitatively, then I think the term should not be attributed to the survey sites in this paper. Perhaps this could be suggested as an area of future research.

Minor comments:

Line 12: I agree that electrocution likely impacts bat populations, but my understanding is that the data needed to show that declines are due to electrocution are missing. I would therefore revise by inserting “likely” before “affecting bat populations.”

Line 19-20 and in results: Given that the range is 0-24.6 electrocutions per km (for all power line types) and the mean is 0.92 electrocutions per km for 4V lines, suggests that the distribution is skewed or that the value of 24.6 is an outlier. I would like to see the distributions for each of the power line types. How do they compare to each other? And within each type, how does the mean compare to the median?

Line 22-23: Revise as “…new power lines and old vertically-oriented power wires with high rates of electrocution.” Unless hotspot can be quantified, this term should not be used.”

Line 24: Given that this manuscript does not provide census data to support this claim, I would revise sentence as “…not only may contribute to their population decline, but also put their ecosystems services at risk.”

Line 50: “on” should be “in gray literature”

Line 51: Replace “On the one hand” with “First”

Line 53: Replace “On the other hand” with “Second”

Line 64-72: Text is a mix of results and discussion. I think it should be replaced with a succinct description of the types of data that were collected during the surveys to assess electrocution risk to Indian flying foxes. Specifically, what risk factors were thought to be important?

Line 82-84: How exactly did you identify individual powerlines? For example, where does each one start and stop? Given that power lines are all connected, I would think the authors had to make many decisions about this issue. More details about this process and the criteria used are needed to make the survey method repeatable by other individuals.

Line 98: Authors state that 4 people were looking for carcasses. Does this include the driver? Seems risky to me.

Line 99: Remove “easily” as I don’t see how this was quantified in the study.

Line 129-135: Report the sample size for the number of occasions in which this occurred. It seems like you could simply use a paired t-test to determine if there was a difference in electrocutions between the pairs of 4V and H lines and the pairs of 3V and H lines. Later on in the results (line 180) you state that the sample size is insufficient for statistical testing, but 16 pairs is sufficient.

Line 139: Not clear what you mean by “gathering”

Line 158: Radius is misspelled in the Table title. What is the order of the tree species in the table? Are they arranged phylogenetically? Perhaps reorganizing by importance or by status (exotic vs. native, etc) and indicating status would be helpful to the reader. I would also add a total row at the bottom or indicate the total number of trees in the Table title.

Line 184: Length is misspelled in the table

Line 221: days should be single

Line 222: Revise “a colony” with “one colony

Line 224: Not clear why you state that the actual number of electrocutions were surely much higher here. Likely because not all carcasses are seen by searchers and because some carcasses are removed by scavengers before they are found. This should be explained.

Line 269-270: Revise as “For example, research dealing…in recent years and is increasingly focused on developing effective mitigation and minimization strategies.” Could add this reference: American Wind Wildlife Institute (AWWI). 2018. Bats and Wind Energy: Impacts, Mitigation, and Tradeoffs. Washington, DC. Available at www.awwi.org

Line 272: Need to add the caveat that you only looked for flying foxes hanging on wires. Searches for carcasses beneath wires could be conducted to confirm that this is the case.

Line 277: Substitutes is misspelled

Line 286-287: Pteropus scapulatus should be italicized

Line 289: By “oriented” do you mean targeted or focused? I would use one of these terms instead.

Line 293: Replace “after divulgating…” with “once they are made aware of the decreasing population trends and key ecosystem services provided by these animals.”

Line 296-298: Same comments as above. You would need to have conducted surveys beneath the wires to definitely show that H wires have no impact. This statement should be softened.

Line 311: do you mean “and in other countries.”?

Author Response

I have carefully read the article “Urban sprawl, food subsidies, and power lines: an ecological trap for large frugivorous bats in Sri Lanka” submitted by Tella and co-authors. The manuscript is well written and easy to follow, providing important information on the extent to which Indian flying foxes are electrocuted at power lines in southern Sri Lanka. I think the manuscript will be of interest to the readers of Diversity and, if published, may help shed light on the importance of this issue and garner support for implementing mitigation strategies. I have two major comments and several minor questions/comments for the authors.

Thank you very much for the careful review, positive comments, and questions/corrections provided below.

There needs to be some mention of how detectability of electrocuted flying foxes could vary among the power lines types, as well as among the 3 landscape types, and how this would influence the estimates of electrocution. For example, the authors state that electrocuted flying foxes wouldn’t be found on the ground below horizontal wires, but it appears that the authors did not substantiate this claim (i.e., they did not conduct ground surveys under any wire type). I think the authors’ explanations make sense (that flying foxes get electrocuted when they roost on 4V and 3V wires and therefore can contact 2 wires at once, and that electrocution risk is much less when they roost on H wires), but this should be verified. It would also be helpful to have some idea about how long carcasses persist in the wire and are therefore able to be detected – is it one or two days or can it be much longer? Are the wires equally visible by human observes along the roads? I would imagine that wires could be obscured by buildings in the more urban areas and obscured by trees and other vegetation in the suburban or agricultural areas. Although the authors don’t appear to have data on detectability, this is an important issue that should be addressed in the manuscript.

We feel doubts about detectability could arise from our figure 1f, where the electrocuted flying fox looks like a small point with vegetation behind it. However, electrocuted flying foxes were easily detectable, given their big size (wingspan 115-130 cm, body mass 0.8 – 1.4 kg) and that all the surveyed power lines were along the margins of the roads/streets surveyed by slowly driving a car, running in parallel and very close (usually 2-5 m) to the trajectory of the car. To illustrate it with a few examples, the pictures shown below were took from the car. As the close proximity of power lines to the road was true for all habitats, we did not find evidence for differences on this –high- detectability among habitats. We have clarified this issue in Methods. 

Regarding ground surveys, we have reworded that sentence in Discussion for clarity.  Our aim (now better stated in Introduction) was to conduct a baseline survey at a large spatial scale, resulting in > 900 km of power lines surveyed. A ground survey could be not conducted at such a large spatial scale. Moreover, a small-scale ground survey could give biased results depending of the locality sampled. In fact, to our knowledge the only published ground survey was conducted in a locality with an extreme mortality rate (74 electrocuted flying foxes in 3 km of power line, i.e. 24.7 indiv./km, ref. 22). Surprisingly, all the electrocuted individuals were found hanging on wires but none on the ground, and thus the authors of this study suspected that it could result from the action of dogs and other scavenging animals removing the carcasses soon after falling to the ground (ref. 22). Therefore, as we recognized in Discussion, we surely underestimated the number of electrocuted individuals as an unknown proportion of carcasses could fall to the ground and thus be not recorded even by ground surveys. Consequently, among other research needs we have added in section 4.3, last paragraph, the need of conducting more detailed studies following our baseline survey, to estimate reliable daily mortality rates taking into account the time carcasses remain hanging on wires and later on after falling to the ground.

The authors mention hotspots of fatality as being particularly important, in part for their potential to be strategically mitigated, but their analyses do not appear to reveal any hotspots. In line 182, they state that differences in mortality are not due to any particular spatial location. Is there another analytical approach that could be used to detect “hotspots” in the data? If hotspots cannot be identified quantitatively, then I think the term should not be attributed to the survey sites in this paper. Perhaps this could be suggested as an area of future research.

We highlight our baseline survey was not aimed to identify mortality hotspots, but to obtain a first approach to the large-scale impact of electrocution on Indian flying foxes. From our descriptive analyses of the spatial location of fatalities, it seems mortality is unrelated to the distance to colonies, and we discussed the potential caveats of this result. However, it does not discard the existence of hotspots. Rather, the existence of particular power lines with high mortality rates suggests there are hotspots. As already discussed, the identification of mortality hotspots would aid to optimize mitigation efforts and resources. We already suggested that citizen science could help to locate power lines with high electrocution incidences. Moreover, we have now added in section 4.3, as an area of further research, the need to identify mortality hotspots citing a new reference (ref. 47) that provides methodological and statistical advice to properly identify them, once more accurate mortality rates will be obtained (see our previous comment, above).  

 Minor comments:

Line 12: I agree that electrocution likely impacts bat populations, but my understanding is that the data needed to show that declines are due to electrocution are missing. I would therefore revise by inserting “likely” before “affecting bat populations.”

Done

Line 19-20 and in results: Given that the range is 0-24.6 electrocutions per km (for all power line types) and the mean is 0.92 electrocutions per km for 4V lines, suggests that the distribution is skewed or that the value of 24.6 is an outlier. I would like to see the distributions for each of the power line types. How do they compare to each other? And within each type, how does the mean compare to the median?

Here is a figure showing the distribution of mortality rates for each power line type (yellow: H, red: 3V, green: 4V):

The distribution is skewed following a Poisson distribution, as it is expected for fatalities in linear infrastructures such as power lines (see new reference 47). We have clarified it in Methods. It is true there is a gap between 7 and 16.67 electrocutions/km. However, when excluding the two extreme values (16.67 and 24.62 electrocutions/km) from the GLM the results are nearly identical (see table below and compare it with Table 2 in the manuscript). Therefore, results are not influenced by these two values and we have maintained the original analysis in the text, although indicating that results remain the same when excluding them.

Line 22-23: Revise as “…new power lines and old vertically-oriented power wires with high rates of electrocution.” Unless hotspot can be quantified, this term should not be used.”

See response above

Line 24: Given that this manuscript does not provide census data to support this claim, I would revise sentence as “…not only may contribute to their population decline, but also put their ecosystems services at risk.”

Done

Line 50: “on” should be “in gray literature”

Done

Line 51: Replace “On the one hand” with “First”

Done

Line 53: Replace “On the other hand” with “Second”

Done

Line 64-72: Text is a mix of results and discussion. I think it should be replaced with a succinct description of the types of data that were collected during the surveys to assess electrocution risk to Indian flying foxes. Specifically, what risk factors were thought to be important?

Accordingly, we have reworded it to mention the variables measured and that we related to the spatial distribution of electrocutions. However, we have maintained the structure of this paragraph, briefly commenting our results and conclusions, following the indications of this journal in its template for preparing the Introduction of the manuscripts. It states: : “Finally, briefly mention the main aim of the work and highlight the principal conclusions”

Line 82-84: How exactly did you identify individual powerlines? For example, where does each one start and stop? Given that power lines are all connected, I would think the authors had to make many decisions about this issue. More details about this process and the criteria used are needed to make the survey method repeatable by other individuals.

During the roadside survey, we recorded all electrocuted individuals (n = 300) and related their spatial location to the degree of urbanization and distance to the nearest colony. However, to obtain mortality rates (number of electrocutions/km) related to power line types we only used the individuals (n = 266) found in power lines from which we clearly identified where they start and where they finish (n = 352 power lines), thus allowing to measure their lengths and be fully surveyed. We preferred to only use fully surveyed power lines to calculate mortality rates, at the cost of slightly losing sample sizes. We have clarified it in Methods.

Line 98: Authors state that 4 people were looking for carcasses. Does this include the driver? Seems risky to me.

This has been clarified as follows:Therefore, four people (three observers and the driver)….”

Line 99: Remove “easily” as I don’t see how this was quantified in the study.

These sentences have been reworded (see our first response about detectability)

Line 129-135: Report the sample size for the number of occasions in which this occurred.

These sample sizes were provided in Results, but we have also included them here as required.

It seems like you could simply use a paired t-test to determine if there was a difference in electrocutions between the pairs of 4V and H lines and the pairs of 3V and H lines. Later on in the results (line 180) you state that the sample size is insufficient for statistical testing, but 16 pairs is sufficient.

We could not use a simple paired-t test given the Poisson distribution of data. Therefore, we properly used a GLM with a Poisson distribution, being also consistent with the previous analysis of all electrocutions together. Unfortunately, the GLM restricting data to those 3V and H power lines that coincided in space did not converge due to the small sample size (n = 16 power lines). We have clarified it in the text.

Line 139: Not clear what you mean by “gathering”

We have reworded this sentence, according to the suggestion provided by reviewer 3

Line 158: Radius is misspelled in the Table title.

Thank you for noting this and other typos. We have noticed the template provided by the journal to prepare the manuscript dos not highlight typos.

What is the order of the tree species in the table? Are they arranged phylogenetically? Perhaps reorganizing by importance or by status (exotic vs. native, etc) and indicating status would be helpful to the reader. I would also add a total row at the bottom or indicate the total number of trees in the Table title.

We have rearranged the table by importance (%) of each tree species, and indicated the total number of trees in the legend to the table (note this number was also provided in the text, two lines above). There is only one native species, as reported in the text, so we feel it is not necessary to include an additional column for status.

Line 184: Length is misspelled in the table

Thanks, corrected

Line 221: days should be single

Done

Line 222: Revise “a colony” with “one colony

Done

Line 224: Not clear why you state that the actual number of electrocutions were surely much higher here. Likely because not all carcasses are seen by searchers and because some carcasses are removed by scavengers before they are found. This should be explained.

This sentence has been reworded, see our first response above.

Line 269-270: Revise as “For example, research dealing…in recent years and is increasingly focused on developing effective mitigation and minimization strategies.” Could add this reference: American Wind Wildlife Institute (AWWI). 2018. Bats and Wind Energy: Impacts, Mitigation, and Tradeoffs. Washington, DC. Available at www.awwi.org

Thanks, we have reworded this sentence and added the reference, as well as another relevant one recently published in Diversity.

Line 272: Need to add the caveat that you only looked for flying foxes hanging on wires. Searches for carcasses beneath wires could be conducted to confirm that this is the case.

See our first response above

Line 277: Substitutes is misspelled

Done

Line 286-287: Pteropus scapulatus should be italicized

Done

Line 289: By “oriented” do you mean targeted or focused? I would use one of these terms instead.

We have used now focused

Line 293: Replace “after divulgating…” with “once they are made aware of the decreasing population trends and key ecosystem services provided by these animals.”

Thanks, we have reworded the sentence as suggested

Line 296-298: Same comments as above. You would need to have conducted surveys beneath the wires to definitely show that H wires have no impact. This statement should be softened.

We have softened the sentence as follows: “power lines with wires oriented horizontally (Figure 1e) cause comparatively little impact,…”

As discussed above, and in the manuscript, looking for carcasses beneath the wires (and in the absence of scavengers removing them) would surely increase the mortality rates estimates, but also for V3 and V4 power lines

Line 311: do you mean “and in other countries.”?

Yes, corrected

Reviewer 2 Report

Very interesting paper on my mind.

Minor revisions

line 31 :  The conservation status is very depending on geographical area considered. Then, the conservation status is not always poor. “

lines 153 to 157: Why do you analyse number of electocuted bats depending on exotic or native trees? What is the hypothesis? Whithout considering bats, which the ratio of nature/exotic trees in gardens?

Line 195: about “predator-free” refuges, could you exclude domestic animals as predators?

Line 221: About “we found 300 electrocuted individuals over just a 19-days survey period….” The 1.5% should vary a lot depending on period of fieldwork.

Line 299: regulations should impose horizontal wires but if it is true for bats, we could think that for birds, electrocution should be fewer on vertical wires. Why id you not recommand longer distance between wires of power lines?

Major revisions

Materials ands methods: I think some details or explanations are missing. Why did you choose a period between 10th and 28th to conducted fieldwork? Because of bats activities?

How to be sure that dead bat were electrocuted? You identified them on ground? From how many meters to wires?

In results lines 162 to 164, there is a distinction between systematic survey and a non-systematic one. There is no description of thos non-systematic survey in materials and methods. We do not know where and when it was conducted? And who?

Author Response

Very interesting paper on my mind.

Thank you very much for the positive view and suggestions/comments provided below.

Minor revisions

line 31 :  The conservation status is very depending on geographical area considered. Then, the conservation status is not always poor. “

We fully agree, the conservation status of any species may vary geographically. However, this sentence summarizes the most recent assessments provided by the IUCN Red List, which evaluates the conservation status of the species at a global scale.

lines 153 to 157: Why do you analyse number of electocuted bats depending on exotic or native trees? What is the hypothesis? Whithout considering bats, which the ratio of nature/exotic trees in gardens?

There is not a hypothesis behind these descriptive results. We simply recorded the species of fruiting trees close to the electrocuted bats, finding that almost all were exotic species. As we recorded all the trees within a 30 m radius (that is, not only the closest tree), we feel our survey reflects well the ratio native/exotic species available in the gardens. These results are important, as they suggest that foraging flying foxes were attracted by exotic fruiting trees, thus losing their ecological functions in native forests as we later discuss in the Discussion section.

Line 195: about “predator-free” refuges, could you exclude domestic animals as predators?

No, domestic animals and even humans may act as predators in some cities. In fact, cats are yearly killing thousands of birds in several European and US cities. However, it seems not to be the case in our study area, where cats were very scarce while the very abundant domestic dogs cannot climb trees.

Line 221: About “we found 300 electrocuted individuals over just a 19-days survey period….” The 1.5% should vary a lot depending on period of fieldwork.

We agree this is a roughly calculated rate derived from our baseline survey, as recognized few lines later. We thus have added some sentences in section 4.3, last paragraph, highlighting the need of further research to properly estimate daily mortality rates. 

Line 299: regulations should impose horizontal wires but if it is true for bats, we could think that for birds, electrocution should be fewer on vertical wires.

Regarding birds, we only recorded one house crow (Corvus splendens) electrocuted in our survey, despite this species is widespread and very abundant across the study area, and it occurred in vertical wires. Therefore, electrocution risk seems to be low for birds. We have included this information in section 4.2, end of first paragraph.

Why id you not recommand longer distance between wires of power lines?

We already discussed it in section 4.3, third paragraph. Increasing the distance between wires may be effective for small-sized bat species but it seems unrealistic for flying foxes given their long wingspan (c. 1.2 m). Therefore, a safer distance between wires (e.g., 1.5 m) would force to substitute all existing poles by poles c. 4 m taller, which would make the construction of overhead medium-voltage power lines impractical. Oo et al. (2017) argued the same when discussing mitigation measures to avoid electrocution for flying foxes. We have reworded this sentence, also citing this reference, to make clearer the argument.

Major revisions

Materials ands methods: I think some details or explanations are missing. Why did you choose a period between 10th and 28th to conducted fieldwork? Because of bats activities?

The fieldwork period was just imposed by the time we had available to conduct this baseline survey. There is the possibility that mortality rates could vary across seasons in the case flying foxes would change their activities seasonally. Therefore, we have added the need of further research to fill this knowledge gap (section 4.2, last paragraph).

How to be sure that dead bat were electrocuted? You identified them on ground? From how many meters to wires?

We have clarified in the Methods section that the power lines surveyed were very close (2 -5 m) to the roads and streets we drove to conduct the mortality survey, so it is was very easy to record from the car the very large flying foxes electrocuted. All individuals were clearly electrocuted as they were observed dead and touching two wires. We paste below some pictures showing it, which are similar to all the others published when recording electrocuted flying foxes elsewhere.  

In results lines 162 to 164, there is a distinction between systematic survey and a non-systematic one. There is no description of thos non-systematic survey in materials and methods. We do not know where and when it was conducted? And who?

We are sorry these points were not clearly explained. All data was collected during the same fieldwork period and by the same research team. We have removed the terms systematic and non-systematic surveys, as they are confusing.

During the roadside survey, we recorded all electrocuted individuals (n = 300) and related their spatial location to the degree of urbanization and distance to the nearest colony. However, to obtain mortality rates (number of electrocutions/km) related to power line types we only used the individuals (n = 266) found in power lines from which we clearly identified where they start and where they finish (n = 352 power lines), thus allowing to measure their lengths and be fully surveyed (this is what we referred as “systematic survey”). We preferred to only use fully surveyed power lines to calculate mortality rates, at the cost of slightly losing sample sizes. We have clarified it in Methods and Results.

Reviewer 3 Report

This is an interesting and timely study that contributes valuable data on a yet little researched threat faced by many flying fox populations. Overall, the manuscript is well-written and the ideas are clearly presented. The methods used are sound and the results are clearly presented. I particularly liked the focus of the discussion. I congratulate the authors on this nice contribution, which I hope will trigger more studies in the future on the issue of electrocution. Below I detail a number of minor issues and typos the authors should attend to in a revision of the manuscript.

l. 20: being highest instead of being maximum

l. 22/23: I suggest rewriting to: ….and old vertically-oriented lines should in electrocution hotspots should be substituted

l. 30: stand out

l. 34: Although much more research is needed…

l. 49: flying fox species

l. 50: in the gray literature

l. 108: instead of saying “for a good subsample” I would simply indicate the sample size here. Also, it would be good to add a bit more detail here about the spatial coverage of this subsample relative to the total number of observations and whether this subsample covered all three main habitat categories.

l. 128: power line as a covariate: do you mean as an offset term?

l. 139: “… within the same study area (Figure 2), gathering each one between 50 and 9,250 individuals.” I suggest rephrasing to: “… within the same study area (Figure 2), each comprised of between 50 and 9,250 individuals.”

l. 141: please also include the SD in addition to the mean, here and elsewhere

l. 146: So all colonies were in the habitat category “urban” or did you also find colonies in suburban and agricultural areas?

l. 155: typo – trees

l. 156: 4.77m – probably unnecessary level of precision, applies also to other figures reported elsewhere

l. 162: Do the numbers referred to in Figure 3 and section 3.1. relate to only those 266 recorded during the systematic surveys or the total of 300?

l. 176: simultaneously

l. 180: mortality rates

l. 233: as a response to

l. 242: flying fox colonies

l. 251: trees

l. 277: substitutes

l. 286: italicize species name

l. 289: platform

l. 297: flying foxes

l. 298: there are no other wires…

l. 305: horizontally

l. 313: P. poliocephalus

l. 315: given that

l. 317: there is available information

Figure 3: please increase font size of axis labels, which in (b) and (c) are barely readable.

Figure 4: use decimal points for values plotted on the y axis, the same applies to those reported in Table 2.

Table 1: typo in legend: radius, not radious. Also, please italicize all species names in the table and use decimal points instead of a comma in the % column.

Author Response

This is an interesting and timely study that contributes valuable data on a yet little researched threat faced by many flying fox populations. Overall, the manuscript is well-written and the ideas are clearly presented. The methods used are sound and the results are clearly presented. I particularly liked the focus of the discussion. I congratulate the authors on this nice contribution, which I hope will trigger more studies in the future on the issue of electrocution. Below I detail a number of minor issues and typos the authors should attend to in a revision of the manuscript.

Thank you very much for the positive comments and for the numerous corrections provided below. We have noticed the journal’s template used to prepare the manuscript does not highlight typos.

  1. 20: being highest instead of being maximum

Done

  1. 22/23: I suggest rewriting to: ….and old vertically-oriented lines should in electrocution hotspots should be substituted

Done

  1. 30: stand out

Done

  1. 34: Although much more research is needed…

Done

  1. 49: flying fox species

Done

  1. 50: in the gray literature

Done

  1. 108: instead of saying “for a good subsample” I would simply indicate the sample size here. Also, it would be good to add a bit more detail here about the spatial coverage of this subsample relative to the total number of observations and whether this subsample covered all three main habitat categories.

Done

  1. 128: power line as a covariate: do you mean as an offset term?

No, an offset term is usually used for a covariate with known slope. It is expected that the number of fatalities will increase with the length of the power line surveyed, but with an unknown slope. Then, fitting power line length as a covariate allows obtaining a coefficient and to control the results for sampling effort (i.e., the length of power lines surveyed).

  1. 139: “… within the same study area (Figure 2), gathering each one between 50 and 9,250 individuals.” I suggest rephrasing to: “… within the same study area (Figure 2), each comprised of between 50 and 9,250 individuals.”

Done

  1. 141: please also include the SD in addition to the mean, here and elsewhere

Done

  1. 146: So all colonies were in the habitat category “urban” or did you also find colonies in suburban and agricultural areas?

All colonies were in urbanized areas, rephrased for clarity

  1. 155: typo – trees

Done

  1. 156: 4.77m – probably unnecessary level of precision, applies also to other figures reported elsewhere

Rounded to one decimal

  1. 162: Do the numbers referred to in Figure 3 and section 3.1. relate to only those 266 recorded during the systematic surveys or the total of 300?

As explained in the text, this section refers to all the electrocutions recorded (n =300). Please note we have removed the term “systematic survey” to avoid confusions.

  1. 176: simultaneously

Done

  1. 180: mortality rates

Done

  1. 233: as a response to

Done

  1. 242: flying fox colonies

Done

  1. 251: trees

Done

  1. 277: substitutes

Done

  1. 286: italicize species name

Done

  1. 289: platform

Done

  1. 297: flying foxes

Done

  1. 298: there are no other wires…

Done

  1. 305: horizontally

Done

  1. 313: P. poliocephalus

Done

  1. 315: given that

Done

  1. 317: there is available information

Done

Figure 3: please increase font size of axis labels, which in (b) and (c) are barely readable.

We think they are readable just after increasing a bite the size of the figure

Figure 4: use decimal points for values plotted on the y axis, the same applies to those reported in Table 2.

Done

Table 1: typo in legend: radius, not radious. Also, please italicize all species names in the table and use decimal points instead of a comma in the % column.

Done

Round 2

Reviewer 2 Report

No more comments and suggestions after the first review.